# LEARNING RICH IMAGE REPRESENTATION WITH DEEP LAYER AGGREGATION

**Fisher Yu, Dequan Wang, Evan Shelhamer & Trevor Darrell**
UC Berkeley

## ABSTRACT

Architectural efforts are exploring many dimensions for network backbones, designing deeper or wider architectures, but how to best aggregate layers and blocks across a network deserves further attention. We augment standard architectures with deeper aggregation to better fuse information across layers. Our deep layer aggregation structures iteratively and hierarchically merge the feature hierarchy to make networks with better accuracy and fewer parameters. Experiments across architectures and tasks show that deep layer aggregation improves recognition and resolution compared to existing branching and merging schemes.

## 1 INTRODUCTION

Representation learning and transfer learning now permeate computer vision as engines of recognition. The simple fundamentals of compositionality and differentiability give rise to an astonishing variety of deep architectures (Krizhevsky et al., 2012; Szegedy et al., 2015; Simonyan & Zisserman, 2015; He et al., 2016; Zagoruyko & Komodakis, 2016). The rise of convolutional networks as the backbone of many visual tasks, ready for different purposes with the right task extensions and data (Girshick et al., 2015; Shelhamer et al., 2016; Xie & Tu, 2015), has made architecture search a central driver in sustaining progress. The ever-increasing size and scope of networks now directs effort into devising design patterns of modules and connectivity patterns that can be assembled systematically. This has yielded networks that are deeper and wider, but what about more closely connected?

In this work, we investigate how to aggregate layers to better fuse semantic and spatial information for recognition and localization. Extending the "shallow" skip connections of current approaches, our aggregation architectures incorporate more depth and sharing. We introduce two structures for deep layer aggregation (DLA): iterative deep aggregation (IDA) and hierarchical deep aggregation (HDA). These structures are expressed through an architectural framework, independent of the choice of backbone, for compatibility with current and future networks. IDA focuses on fusing resolutions and scales while HDA focuses on merging features from all modules and channels. IDA follows the base hierarchy to refine resolution and aggregate scale stage-by-stage. HDA assembles its own hierarchy of tree-structured connections that cross and merge stages to aggregate different levels of representation. Our schemes can be combined to compound improvements.

Our experiments evaluate deep layer aggregation across standard architectures and tasks to extend ResNet (He et al., 2016) and ResNeXt (Xie et al., 2017) for large-scale image classification, fine-grained recognition, semantic segmentation, and boundary detection. Our results show improvements in performance, parameter count, and memory usage over baseline ResNet, ResNeXT, and DenseNet architectures. DLA achieve state-of-the-art results among compact models for classification. Without further architecting, the same networks obtain state-of-the-art results on several fine-grained recognition benchmarks. Recast for structured output by standard techniques, DLA achieves best-in-class accuracy on semantic segmentation of Cityscapes (Cordts et al., 2016) and state-of-the-art boundary detection on PASCAL Boundaries (Premachandran et al., 2017). Deep layer aggregation is a general and effective extension to deep visual architectures.

## 2 DEEP LAYER AGGREGATION

We define aggregation as the combination of different layers throughout a network. In this work we focus on a family of architectures for the effective aggregation of depths, resolutions, and scales. We

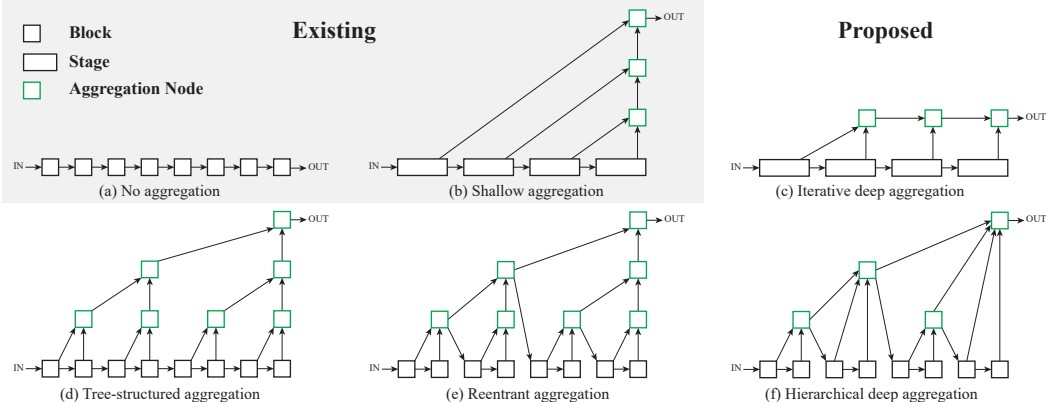

Figure 1: Different approaches to aggregation. (a) composes blocks without aggregation. (b) combines parts of the network with skip connections. We propose two deep aggregation architectures: (c) aggregates iteratively by reordering the skip connections of (b) such that the shallowest parts are aggregated the most for further processing and (d) aggregates hierarchically through a tree structure of blocks. (e) and (f) are refinements of (d) that deepen aggregation by routing intermediate aggregations back into the network and improve efficiency by merging successive aggregations at the same depth. Our experiments show the advantages of (c) and (f) for recognition and resolution.

call a group of aggregations *deep* if it is compositional, nonlinear, and the earliest aggregated layer passes through multiple aggregations.

**Iterative deep aggregation** follows the iterated stacking of the backbone architecture. We divide the stacked blocks of the network into stages according to feature resolution. Deeper stages are more semantic but spatially coarser. Skip connections from shallower to deeper stages merge scales and resolutions. However, the skips in existing work, e.g. FCN (Shelhamer et al., 2016), U-Net (Ronneberger et al., 2015), and FPN (Lin et al., 2017), are linear and aggregate the shallowest layers the least, as shown in Figure 1(b).

We propose to instead progressively aggregate and deepen the representation with IDA. Aggregation begins at the shallowest, smallest scale and then iteratively merges deeper, larger scales. In this way shallow features are refined as they are propagated through different stages of aggregation. Figure 1(c) shows the structure of IDA.

**Hierarchical deep aggregation** merges blocks and stages in a tree to preserve and combine feature channels. With HDA shallower and deeper layers are combined to learn richer combinations that span more of the feature hierarchy. While IDA effectively combines stages, it is insufficient for fusing the many blocks of a network, as it is still only sequential. The deep, branching structure of hierarchical aggregation is shown in Figure 1(d).

Having established the general structure of HDA we can improve its depth and efficiency. Rather than only routing intermediate aggregations further up the tree, we instead feed the output of an aggregation node back into the backbone as the input to the next sub-tree, as shown in Figure 1(e). This propagates the aggregation of all previous blocks instead of the preceding block alone to better preserve features. For efficiency, we merge aggregation nodes of the same depth (combining the parent and left child), as shown in Figure 1(f).

## 3   APPLICATIONS

We now design networks with deep layer aggregation for visual recognition tasks. To study the contribution of the aggregated representation, we focus on linear prediction without further machinery. Our results do without ensembles for recognition and context modeling or dilation for resolution.

Our classification networks augment ResNet and ResNeXT with IDA and HDA. These are staged networks, which group blocks by spatial resolution, with residual connections within each block. The end of every stage halves resolution, giving six stages in total, with the first stage maintaining the

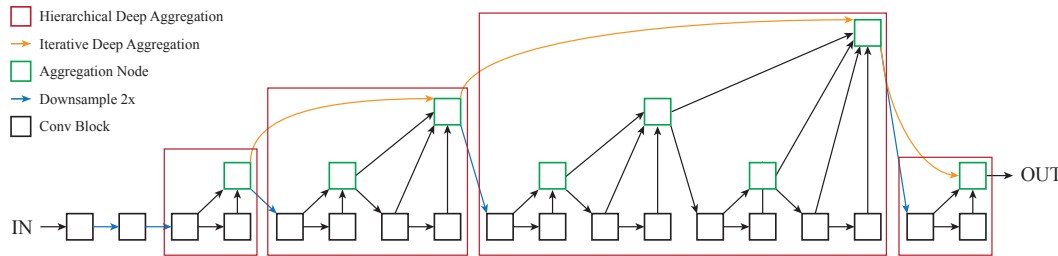

Figure 2: Deep layer aggregation learns to better extract the full spectrum of semantic and spatial information from a network. Iterative connections join neighboring stages to progressively deepen and spatially refine the representation. Hierarchical connections cross stages with trees that span the spectrum of layers to better propagate features and gradients.

input resolution while the last stage is 32× downsampled. The final feature maps are collapsed by global average pooling then linearly scored. The classification is predicted as the softmax over the scores.

We connect across stages with IDA and within and across stages by HDA. These types of aggregation are easily combined by sharing aggregation nodes. In this case, we only need to change the root node at each hierarchy.

For a direct comparison of layers and parameters in different networks, we build networks with a comparable number of layers as ResNet-34, ResNet-50 and ResNet-101. (The exact depth varies as to keep our novel hierarchical structure intact.) To further illustrate the efficiency of DLA for condensing the representation, we make compact networks with fewer parameters. Figure 2 shows a DLA architecture with HDA and IDA.

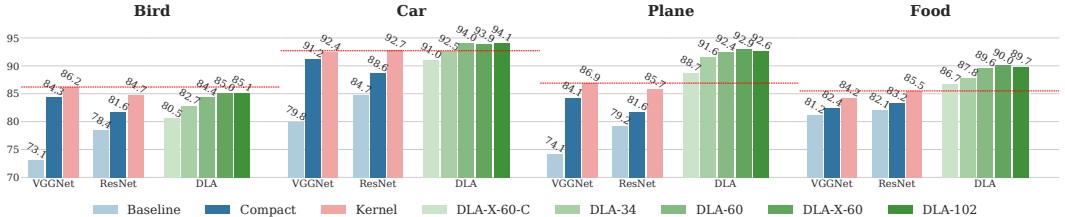

Figure 3: Comparison with state-of-the-art methods on fine-grained datasets. Image classification accuracy on Bird (Wah et al., 2011), Car (Krause et al., 2013), Plane (Maji et al., 2013), and Food (Bossard et al., 2014). Higher is better. $P$ is the number of parameters in each model. For fair comparison, we calculate the number of parameters for 1000-way classification. V- and R- indicate the base model as VGGNet-16 and ResNet-50, respectively. The numbers of Baseline, Compact (Gao et al., 2016) and Kernel (Cui et al., 2017) are directly cited from Cui et al. (2017).

Semantic segmentation, contour detection, and other image-to-image tasks [1] can exploit the aggregation to fuse local and global information. The conversion from classification DLA to fully convolutional DLA is simple and no different than for other architectures. We make use of interpolation and a further augmentation with IDA to reach the necessary output resolution for a task.

IDA for interpolation increases both depth and resolution by projection and upsampling. All the projection and upsampling parameters are learned jointly during the optimization of the network. The upsampling steps are initialized to bilinear interpolation and can then be learned as in Shelhamer et al. (2016). We first project the outputs of stages 3–6 to 32 channels and then interpolate the stages to the same resolution as stage 2. Finally, we iteratively aggregate these stages to learn a deep fusion of low and high level features. Note that we use IDA twice in this case: once to connect stages in the backbone network and again to recover resolution.

---

[1]For further details and experimentation with our model please see the full length version of this paper on arXiv (Yu et al., 2017).

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
