# OpenReview forum: "Learning Rich Image Representation with Deep Layer Aggregation"
_ICLR.cc/2018/Workshop — Accept_

### Official Review · AnonReviewer3 · 2018-02-27
**An interesting contribution not properly formatted for 3 pages**

**Rating:** 5
**Confidence:** 4

**Review:**

This paper presents an interesting study that explores different methods for aggregating features across layers of a deep convolutional network, using ResNet and ResNext as baseline architectures. The aggregations are structured in two different types, iterative (IDA) and hierarchical (HDA) deep aggregations, considering three different variations of the HDA one.

The text is clear and well motivated until reaching the experimental section, where the claims of the text do not match the presented results. While the introduction states that experiments have been run in tasks of large-scale image classification, fine-grained recognition, semantic segmentation and boundary detection, the text only includes a Figure on four datasets for fine-grained classification. Moreover, the Figure is not referred in the text and the legend does not match the text (eg. refers to VGG, to configurations -60 and -102, or in the text they talk about parameter P). Another missing baseline in the results is DenseNet, which is mentioned in the introduction. It seems this workshop paper is a shortened version of a longer paper published on arXiv and pointed by the authors in a footnote, and that the reduction of the length was not made with enough care.

If focusing on the reported results, they only address accuracy performance. However, it would also be desirable to include a discussion and/or figures regarding the memory requirements of using aggregated features, as well as a clear statement regarding if the proposed architectures increase the amount of parameters in the model.

PROS
- The significance of the research question is relevant and interesting.
- Authors with a strong scientific reputation claim state of the art on several fine-grain recognition benchmarks, best-in-class accuracy on semantic segmentation in Cityscapes and state-of-the-art boundary detection on PASCAL.
- Sections 1 and 2 are well written an motivated.

CONS
- The excellent results announced in the introduction are not included in the paper, which makes the paper not self-contained. The text should be rewritten to be self-contained. Focusing in a single task and just mentioning the rest at the very end of the text would greatly improve the clarity of the work.
- Reported results do not explicitly compare with any other result from the state of the art (or, at least, it is not clear that this is done according to the content of Figure 3).
- Aggregating features seems to require higher memory and computational requirements, which are not considered in the included results.

---

### Official Review · AnonReviewer2 · 2018-03-02
**Interesting forms of aggregating layers**

**Rating:** 7
**Confidence:** 4

**Review:**

This paper proposes a more complex form of aggregation of layers from different parts of the neural netwwork. In particular, the authors proposes two forms of aggregation that are then jointly used in their final models. The first is called iterative deep aggregation, which sequentailly aggregates different stages of the network. The second is called hierarchical deep aggregation and it merges different blocks within a stage of the network. Authors compare a network enriched with the two aggregations and show improved results in 3 out of 4 fine-grained classification datasets.

Pros:
- The paper is in general well presented, and Fig. 1 helps to understand the idea.
- The idea of generalizing the concept of aggregation is nice, although, in practice there can be so many configuration and details that I am not sure Fig. 1 can really include all forms of aggregation.

Cons:
- Tha authors did not consider Convolutional Neural Fabric [Saxena and Verbeek, NIPS'17], in which the different resolution levels can be though as a form of aggregation.
- The authors should mention/consider also Dense DenseNet [Huang et al. 2016] and FractalNet [Larsson et al. 2016]. The aggregations are done only at stage level (within convolutions at the same resolution), but they are still based on aggregations of layers and they can resemble the iterative aggregation.
- The authors should make clear that HDA is done within one stage, while IDA is done over multiple stages. The difference is somehow shown in Fig. 1, but it took me a bit to understand that.
- The evaluation is limited, but for a workshop paper it can be enough

Additional comments:
It would be interesting to see an ablation study to understand which form of aggregation is the more effective

---

### Official Review · AnonReviewer1 · 2018-03-08
**Effective alternative to skip connections**

**Rating:** 8
**Confidence:** 4

**Review:**

The paper looks at several candidate methods for aggregating information across layers in convolutional networks beyond the usual skip and residual connections. Specifically, it considers a simple iterative aggregation scheme, where each stage is aggregated in a side channel from the backbone network, and several hierarchical structures. Also considered is re-entrant aggregation, where the outputs of the aggregation stages fed back to the backbone network. Experiments demonstrate that the proposed deep layer aggregation scheme can outperform several competing methods on four fine grained classification tasks.

This is essentially a brief, but well-written, summary of a longer arXiv paper, which contains substantially more experiments and details. The workshop paper is a little light on details, but these are readily found in the longer version. One criticism is that the workshop paper is not self-contained: it is not possible to interpret the algorithms referred to in the legend of Fig 3 without reference to the arXiv paper.

Strengths: well-written, novel and sensible alternative to skip connections, extensive experiments (in the arXiv paper), and solid believable conclusions. Should be of broad interest.

Weaknesses: the submitted paper is not self contained. Mainly experimental work; not much in the way of theoretical justification.

---

### Decision · Program_Chairs · 2018-03-20
**ICLR 2018 Workshop Acceptance Decision**

**Decision:**

Accept

**Comment:**

Congratulations, your paper was accepted to the ICLR workshop.